

# A N-dimensional Fortran Interpolation Program (NterGeo.v2020a) for Geophysics Sciences - Application to a back-trajectory program (BACKPLUMES.v2020r1) using CHIMERE or WRF outputs

Bertrand Bessagnet[1,2], Laurent Menut[1], and Maxime Beauchamp[3]

[1]LMD/IPSL, École Polytechnique, Institut Polytechnique de Paris, ENS, PSL Université, Sorbonne Université, CNRS, 91128 Palaiseau, France
[2]Citepa, Technical Reference Center for Air Pollution and Climate Change, 42, rue de Paradis 75010 Paris, France
[3]IMT Atlantique, Lab-STICC UMR CNRS, 655 Avenue du Technopôle, 29280 Plouzané, France

**Correspondence:** Bertrand BESSAGNET (bertrand.bessagnet@lmd.polytechnique.fr)

**Abstract.** An interpolation program coded in Fortran for irregular N-dimensional cases is presented and freely available. Needs of interpolation procedure over irregular meshes or matrixes with interdependent input data dimensions is frequent in geophysical models. Also, these models often embed look-up tables of physics/chemistry modules. Fortran is a powerful and fast language, highly portable and easy to interface with other existing Fortran models. Our program does not need any libraries

and can be compiled with any Fortran compiler. The program is fast and competitive compared to current Python libraries. A novel optional parameter (normalisation option) is provided when considering different types of units on each dimension. For the general program, the inverse distance is used for the weight calculation with a distance defined as a p-distance. Some tests and examples are provided and available in the code package. Moreover, a real case of geophysics application embedding this interpolation program is provided and discussed, it consists in determining back-trajectories using atmospheric dispersion or

mesoscale meteorological model outputs, respectively from the widely used models CHIMERE and WRF.

## 1 Introduction

Interpolation is commonly used in geophysical sciences for post-treatment operations to evaluate model performances at point observations. The NCO library (Zender, 2008) is commonly used in its recent version V4.9.2 to propose horizontal and vertical interpolations to manage climate models outputs. The most frequent need is to interpolate in 3D spatial dimension and time

therefore in 4 dimensions. Fortran is used extensively for weather and climate-related software (Sun and Grimmond (2019), e.g. WRF - Skamarock et al. (2008); GFDL AM3 - Donner et al. (2011)). Geophysical models can use look-up tables of complex modules instead of a full coupling strategy which is the case of the CHIMERE model (Mailler et al., 2017) with the embedded ISORROPIA (Nenes et al., 1998, 1999) module dealing with chemistry and thermodynamics.

In this case the look-up table can easily exceed 5 dimensions to approximate the model. In parallel, Artificial Intelligence

methods are developed and can explore the behaviour of complex model outputs that requires fast interpolation methods. While more recent modern languages like Python are used in the scientific community, Fortran remains widely in the geophysics / engineering community and is known as one of the faster language in time execution, very good on array handling, parallelisa-



tion and above all portability. Some benchmarks are available on web site to evaluate the performances of languages on simple to complex operations (Kouatchou, 2018).

In some studies, the parameterization techniques proposed to manage aerosol/droplet microphysical schemes (Rap et al., 2009) employ either the modified Shepard interpolation method (Shepard, 1968) or the Hardy multiquadrics interpolation method (Hardy, 1971, 1990), and the numerical results obtained show that both methods provide realistic results for a wide range of aerosol mass loadings. For the climate community, comparison of six methods for the interpolation of daily European climate data are proposed by (Hofstra et al., 2008), some of these methods use kriging like methods with the capability to use

co-predictors like the topography.

A python procedure called *scipy.interpolate.griddata* is freely available (Scipy, 2014), unfortunately this program is too general handling fully unstructured datasets and then not enough optimized for our objective. The goal of this paper is to present a program to interpolate in a grid or a matrix which can be irregular (varying intervals) but structured with the possibility to have interdependent dimensions (e.g. a longitude interval edges which depend on longitude, latitude, altitude and time). We

think this type of program can be easily implemented within models or to manage model outputs for post-treatment issues.

Atmospheric models (physics and/or chemistry) are commonly used in the Geophysics community. Among all existing models, HYSPLIT (Stein et al., 2015), STILT (Lin et al., 2003) and its WRF coupled version, (Nehrkorn et al., 2010) and Flexpart (Pisso et al., 2019) are widely used. These models have different levels of complexity and are able to transport backward in time air masses by accounting for atmospheric motions, chemistry and deposition processes.

A new back-trajectory plume has been developed taking advantage of this new interpolation program used to perform a 3D spatial interpolation at each time step. Compared to the other codes some additional characteristics are implemented, the code is very light and does not required a lot of computer resources and libraries. This code is also very fast and enables to calculate numerous trajectories in a few minutes. Finally, and probably the most important point, the code is dedicated to run on the models outputs already calculated: trajectories are thus estimated over exactly the same grid than the one used for the direct

Eulerian simulation. This model is thus particularly suitable for users already having an Eulerian simulation and who want supplementary information about their studied case.

This paper describes (i) the methodology and the content of the interpolation program package **NterGeo**, and (ii) an application of this program embedded in the new "back-trajectory" program **BACKPLUMES**. These two codes are freely available (see code availability section).

## 2   Development of the interpolation program

The program supports the exploration of irregular but structured grids or look-up tables defined by a size in each dimension which can be of course different. The space intervals can vary along a dimension and the grid interval edges in each dimension can depend on other dimensions. Two versions have been developed, (i) a version for a "regular" arrays with independent dimensions and, (ii) a "general" version for possible inter-dependent dimensions, e.g. to handle 3D meshes which have time

varying spatial coordinates. The code does not need any libraries and can be easily compiled with any Fortran compiler. Our



interpolation code was tested with gfortran (GNU Fortran project) and ifort (Intel). As it includes not specific options or function, version of a compiler, there is no reason to have limitations or errors with other compilers. The top shell calling script in the package provide two sets of options for "production" and "debugging" modes. Assuming the $X$ array, the result of the function $f$ transforming $X$ to $Y$ array in $\mathbb{R}$ can be expressed as:

$$Y = f(X(x_1, \ldots, x_N)) \tag{1}$$

$N$ is the dimension of the array, $x_i$ is the coordinates at dimension $i \in [1, N]$ of the point X we want to interpolate.

## 2.1 The program for regular grids

A program $interpolation\_regular.F90$ for regular grids (i.e. with independent dimensions) is available. To handle this type of grids a classical bilinear interpolation is performed. Figure 1 shows the variables for $N = 3$ defined hereafter in the section.

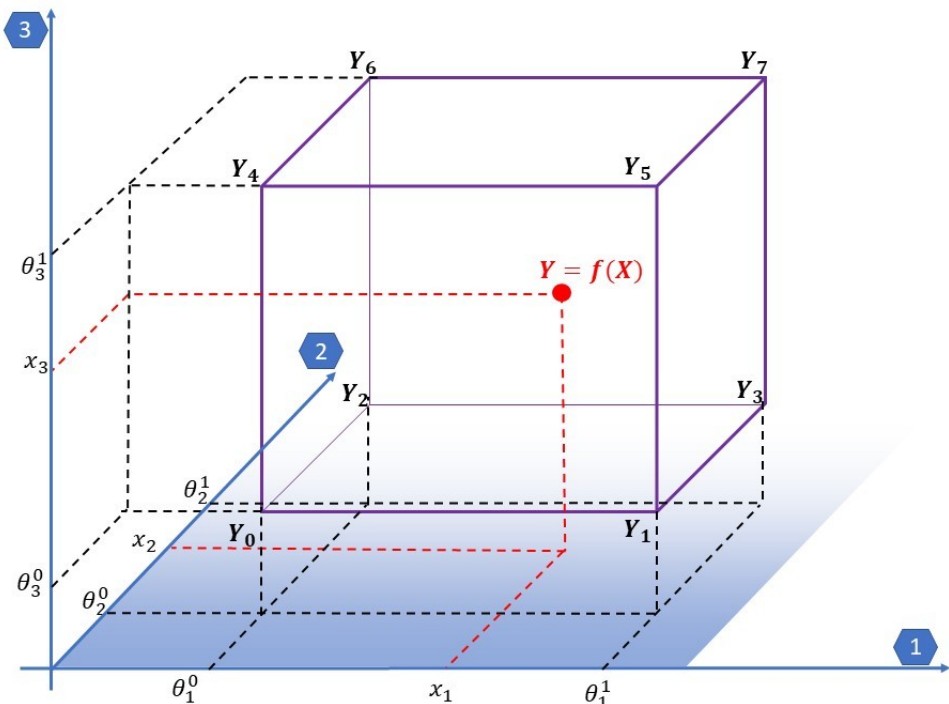

**Figure 1.** *Description of variables for N=3 with a regular grid case*





For the particular case of a regular grid with independent dimensions the result $\tilde{Y}$ of the bilinear interpolation of the $2^N$ identified neighbours can be expressed as:

$$
\begin{aligned}
\tilde{Y} =\, & w_N^0 \ldots w_i^0 \ldots w_1^0 \times Y_0(0 \ldots 0 \ldots 0) \\
& + w_N^0 \ldots w_i^0 \ldots w_1^1 \times Y_1(0 \ldots 0 \ldots 1) \\
& + \ldots \\
& + w_N^{\delta_N} \ldots w_i^{\delta_i} \ldots w_1^{\delta_1} \times Y_k(\delta_N \ldots \delta_i \ldots \delta_1) \\
& + \ldots \\
& + w_N^1 \ldots w_i^1 \ldots w_1^1 \times Y_{2^N-1}(1 \ldots 1 \ldots 1)
\end{aligned}
\tag{2}
$$

with $\delta_i$ the binary digit equal to 0 or 1, and the weights $w_i^{\delta_i}$ for $i \in [1, N]$ defined as:

$$
w_i^0 = \frac{\theta_i^1 - x_i}{\theta_i^1 - \theta_i^0} \qquad\qquad w_i^1 = 1 - w_i^0
\tag{3}
$$

$\Theta_i$ is the list of interval edges on each dimension $i$ and does not depend on other dimensions. $\theta_i^{\delta_i}$ is the bottom ($\delta_i = 0$) and top ($\delta_i = 1$) edges on each dimension $i \in [1 \ldots N]$ so that $x_i \in ]\theta_i^0, \theta_i^1]$. $Y_k$ is a 1-dimensional array with $2^N$ elements storing the value $Y$ of the function at the identified neighbours $\Psi$ on each dimension:

$$
Y_k(\delta_N \ldots \delta_i \ldots \delta_1) = f(\Psi(\theta_N^{\delta_N}, \ldots, \theta_i^{\delta_i}, \ldots, \theta_1^{\delta_1}))
\tag{4}
$$

with $k \in [0, 2^N - 1]$

The tuple $(\delta_N \ldots \delta_i \ldots \delta_1)$ is the binary transformation of integer $k$ defined as $\sum_{i=0}^{N-1}(\delta_i \times 2^i)$. The coefficients $\Gamma_k = w_N^{\delta_N} \ldots w_i^{\delta_i} \ldots w_1^{\delta_1}$ as a product of weighting factors on each direction can be seen as a binary suite that is convenient to handle in a compacted and optimized Fortran programming strategy for the *regular* grid version of the code (Appendix B).

## 2.2   The general program

Considering the *general* program called $interpolation\_general.F90$, the coordinates of edge points are stored in a 1-dimension 
array of $n = \prod_{i=1}^{N} I_i$ elements with $I_i$ the number of edges on each dimension $i$. The tuple of coordinates $(j_1, \ldots, j_N)$ of an interval edge $\theta_k^i$, with $j_i$ the indexed coordinate on dimension $i$, is transformed in a 1-dimension array indexed on $k \in [1, n]$ by:

$$
k = \sum_{k=1}^{N} \left( (i_j - 1) \prod_{l=0}^{j-1} I_l \right) + 1
\tag{5}
$$





with $I_0 = 1$ for initialisation.

Once the nearest neighbour is found, the result $\tilde{Y}$ of the interpolation is a weighting procedure of the $2^N$ closest neighbours using a Shepard interpolation (Shepard, 1968) based on the inverse distance calculations:

$$\tilde{Y} = \sum_{k=0}^{2^N-1} (\Gamma_k \times Y_k) \tag{6}$$

with $Y_k = f(\Upsilon_k)$ the value of the function $f$ at neighbour $\Upsilon_k$ of coordinates $(\theta_k^1, \ldots, \theta_k^N)$, and:

$$\Gamma_k = \frac{1/d_k}{\sum_{k=0}^{2^N-1}(1/d_k)} \tag{7}$$

The distance $d_k$ between the point of interest of coordinates $(x_1, \ldots, x_N)$ to the neighbour $k \in [1, n]$ is calculated as:

$$d_k = \left( \sum_{i=1}^{N} \mid x_i - \theta_k^i \mid^p \right)^{\frac{1}{p}} \tag{8}$$

The previous formula are valid for $d_k \neq 0$, in the case of $d_k = 0$ the procedure stops and exit returning the exact value of the corresponding data of the nearest neighbour. For distorted meshed or matrix, or dimensions with different units (e.g mixing time with length), an hard coded option $norm = .true.$ or $.false.$ is also available to normalize the intervals with an average 95 interval $\Delta_i$ value for the calculation of distances so that:

$$d_k = \left( \sum_{i=1}^{N} \left( \frac{\mid x_i - \theta_k^i \mid}{\mid \Delta_i \mid} \right)^p \right)^{\frac{1}{p}} \tag{9}$$

## 3 Computation strategy for the general program

The list of input/output arguments is provided in Appendix C. In the main program calling the subroutine the key point is to transform first the N-dimension matrix in a 1D array. An example of main program is provided in the code package. The 100 computation strategy in the subroutine can be broken down into the sequential steps as follows:

(I) Find the nearest neighbour of the input data by minimizing a distance with a simple incremental method stepping every $\pm 1$ coordinates on each dimension (detailed later in this section).

(II) Scan the surroundings of the nearest point within the matrix on $\pm 1$ step on each dimension and store the corresponding block of input data to be tested. The size of the block is therefore $(1 + 2 \times 1)^N$ but can be extended to $(1 + 2 \times 2)^N$ if 105 we increase the scanning process to $\pm 2$ on each dimension (hard coded option $iconf$=1 or 2 in the declaration block).

(III) Calculate the distance to the previously selected input data. A $p$-distance concept is adopted (hard coded option $pnum$ in the declaration block). The $pnum$ value $p$ should be superior or equal to 1 to verify the Minkowski inequality and be considered as a metric.





(IV) Sort the previous block of data in ascending order and stop the sorting process when the first $2^N$ point are selected. The code offers the possibility to use only the first $N + 1$ neighbours (hard coded option $neighb$ in the declaration block) that is sufficient and faster in most cases.

(V) Calculate the weights and then the final result.

The first step consisting in finding the first neighbour is the trickiest and is broken down into several steps. Figure 2 displays an example in 2D of the step by step procedure to find the nearest neighbour.

(i) The procedure initializes the process starting from the first point of the input data grid or taken from the last closest point if given in argument as a non-null value.

(ii) A delta of coordinates is applied based on an average delta on each dimension to improve the initialisation. This computation step of delta is externalized as it can be time comsuming and should be done once for all taget points at which we want to interpolate.

(iii) A test between the target value and the input data grid points coordinates determines the $\pm 1$ steps to add on each dimension (see Figure 2 for an example in 2D).

(iv) If the grid point falls on the edges or outside the borders the closest coordinates within the matrix is selected.

(v) A test on the p-distance computation between the running point and the target is performed so that if the distance calculated at Iteration $N_{it}$ is equal to distance at Iteration $N_{it} - 2$ the closest point is found.

(vi) If the distance is too high compared to the characteristic distance of the cell, the point is considered to be outside the borders of the input grid data. Therefore, the code allows a slight extrapolation if the target point is not too far from the borders.

(vii) At this stage, the procedure can stop if the distance to the closest neighbour is 0 returning to the main program with the exact value of the input data grid.

## 4 Visual example in 2D for a regular grid

As an example to visualize the capacity of the *general* program, the 2D function used in Scipy (2014) is used to test our procedure. The function is:

$$Y = f(X) = x_1 \times (1 - x_1) \times cos(4\pi x_1) \times sin(4\pi x_2^2)^2 \tag{10}$$

with $x_1, x_2 \in [0, 1]$.

Our input grid data is a regular grid with regular intervals of 0.02 from 0 to 1 for $x_1$ and $x_2$ with therefore 51 points on



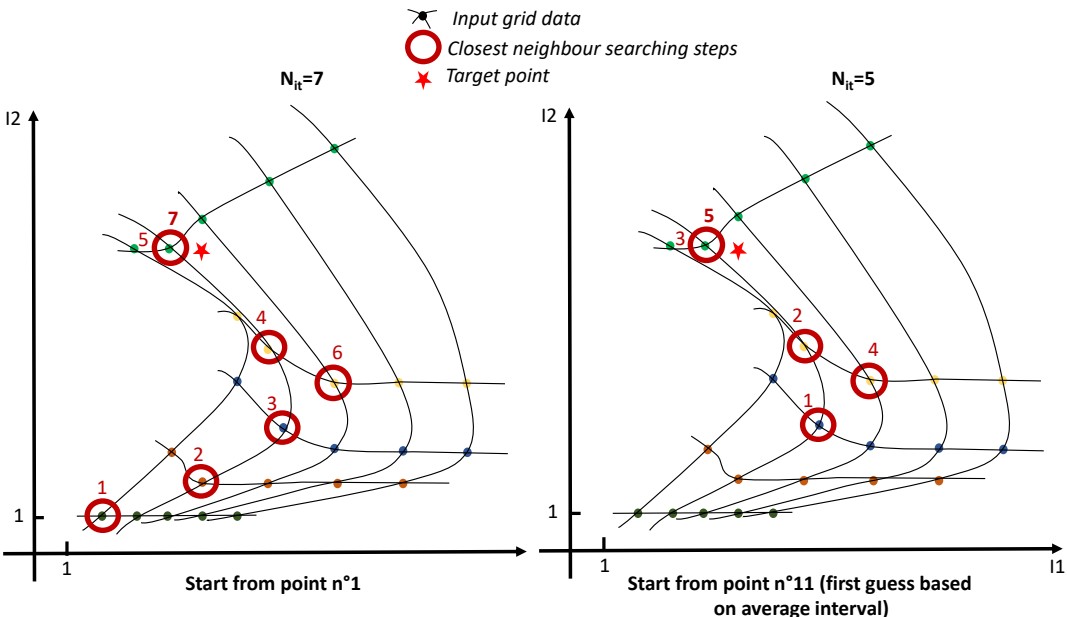

**Figure 2.** *Real example in 2D of the step by step procedure to find the nearest neighbour of a target point for an irregular but structured $5 \times 5$ grid. Left panel when starting the process from the $1^{st}$ point of the grid on the lowest left corner, Right panel when starting with a first guess based on an average delta computed for each dimension.*

each dimension. We propose to interpolate on a finer regular grid with $n$=100×100, 200×200 and 300×300 points on each dimension. For these three interpolations cases a normalized root mean square error (NMSE) of the result $\tilde{Y}_j$ for the full grid point number $j$ can be calculated against the true value $Y_j$ of the function as:

$$NMSE = \frac{\frac{1}{n}\sum\limits_{j=1}^{n}\left(\tilde{Y}_j - Y_j\right)^2}{\frac{1}{n-1}\sum\limits_{j=1}^{n}\left(Y_j - \bar{Y}_j\right)^2} \qquad (11)$$

with $\bar{Y}_j$ the mean value $Y_j$ as $\frac{1}{n}\sum_{j=1}^{n} Y_j$.

For the three cases the CPU time for the interpolation is evaluated and displayed in Table 1 for Machine 1 (Appendix E). The time consuming is somehow proportional to the number of points in which to interpolate. Figure 3 displays the evolution of the NMSE with the parameter $p$ of the p-distance definition. There is a discontinuity of the NMSE from $p=1$ to $p=1^+$ with a slight increase with $p$ in an asymptotic way. The NMSE decreases with the number of points but a slight increase is observed
from 200×200 from 300×300.





**Figure 3.** *Interpolation results for the three cases. Figures generated with the Generic Mapping Tools (Wessel et al., 2019)*

**Table 1.** Performance for each case with $p = 1$

| Case | $100 \times 100$ | $200 \times 200$ | $300 \times 300$ |
| --- | --- | --- | --- |
| NMSE (%) | 0.324 | 0.319 | 0.319 |
| CPU time (s) | 0.45 | 1.84 | 4.1 |



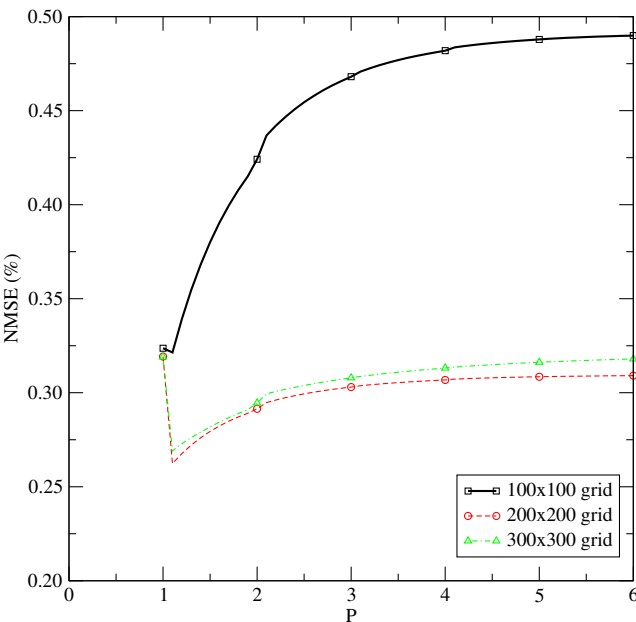

**Figure 4.** *Evolution of performances based on the NMSE for the three cases as a function of the parameter p of the p-distance computation*

## 5 Example in 5D for a regular grid

Still using the *general* program, an example in 5D ($N = 5$) is proposed using the function :

$$Y = f(X) = x_1 \times (1 - x_1) \times cos(4\pi x_1) \times sin(4\pi x_2)$$
$$\times cos(4\pi x_3) \times sin(4\pi x_4) \times cos(4\pi x_5) \tag{12}$$

with $x_1, x_2, x_3, x_4, x_5 \in [0,1]$. The input data grid is a regular grid of $I_i = 35$ interval edges on each dimension $i \in [1,5]$ then

$35^5 = 52\,521\,875$ grid points. The goal is to find the results on a coarse grid of 9 elements on each dimension then $9^5 = 59\,049$ grid points. This case is an opportunity to test the influence of the number of neighbours to calculate the result. In our case, the parameter $p$ of the $p$-distance is set to $p = 2$. The interpolation seems to provide a better performance on the NMSE for our function with less neighbours (case $N + 1$) and obviously with a lower CPU time. This could certainly depends on the type of function to interpolate.

**Table 2.** Performance for the 5D (N=5) case with $p = 2$

| Number of neighbours | $2^N$ | $N + 1$ |
|---|---|---|
| NMSE (%) | 1.570 | 0.870 |
| CPU time (s) | 17.32 | 6.00 |





Another test with the 5D case is performed to test the influence of the normalisation as defined in Equation 9 (flag $norm$) by defining an irregular grid with still $35^5 = 52\,521\,875$ input data points but with (i) random intervals values, and (ii) one dimension depending on another. The definition of the input grid is defined in Appendix D and provided in the code package. With a similar order of magnitude of consumed CPU time the normalisation $norm = .True.$ produce a $NMSE = 0.499\%$ compared to $NMSE = 0.822\%$ for $norm = .False..$ There is then and added value of using such a normalisation with

comparable CPU time consuming (rising from 2.68 to 3.44 s for our case).

## 6   Comparison with Python for a regular grid

The code has been tested against the *Python* procedure *scipy.interpolate.griddata* freely available by (Scipy, 2014), for the following function:

$$
\begin{aligned}
Y = f(X) = & x_1 \times (1 - x_1) \times cos(4\pi x_1) \\
& \times sin(4\pi x_2) \times cos(4\pi x_3)
\end{aligned}
\tag{13}
$$

with $x_1, x_2, x_3 \in [0,1]$. The input data grid is a regular grid of $I_i = 35$ interval edges on each dimension $i \in [1,5]$ then $35^3 = 42875$ grid points. The goal is to find the results on a coarse grid of 9 elements on each dimension then $9^3 = 729$ grid points. A case in 3D has been used for this test because the *Python* library was not able to work with very large datasets (*overflow error*) while our program could make it. $scipy.interpolate.griddata$ is used with the bilinear interpolation option while our method is configured with $p = 2$.

Table 3 clearly shows how the Fortran code is faster compared to the Python library. However the bilinear interpolation method seems to provide a higher accuracy than the inverse distance one embedded in our method. Nevertheless, the error produced by our method looks acceptable.

**Table 3.** Comparison of performances between our method for a 3D case with the gridata Python library. The Machine 2 is used (Appendix F).

|  | Our code with $N+1$ neighbours | Our code with $2^N$ neighbours | Python |
|---|---|---|---|
| NMSE (%) | 0.627 | 1.03 | 0.326 |
| CPU time (s) | 0.04 | 0.04 | 19.49 |

## 7   Geophysics application

### 7.1   Methodology

The back-trajectory module called BACKPLUMES detailed in this section can use output files from CHIMERE (Mailler et al., 2017) or WRF (Skamarock et al., 2008). This kind of model has mainly one calculation to do several times i.e. interpolate the





position of a point in a gridded 3D-times domain. It is why the implementation of a robust and precise interpolation scheme as the one presented in this study is a key point. The first advantage of BACKPLUMES is to use the results of a simulation already performed. The second advantage is to be homogeneous with the "direct" simulation by using the same wind field and the same grid. BACKPLUMES is different than other back-trajectories models, such as Hysplit or Flexpart. Its goal is not to estimate the most likely trajectory as an enveloppe of numerous possible trajectories. Since it is difficult to calculate correct probabilities back in time, the choice was made to randomly launch numerous trajectories and try to cover all possible origins.

The model is easy to use and light because a small set of meteorological parameters is required. These meteorological parameters are described in Table 4 for WRF and CHIMERE. The BACKPLUMES model is an open-source code and is available on the CHIMERE model web site.

| Parameter | Model variable name |
|---|---|
| **WRF model** | |
| Longitude, latitude | XLONG, XLAT |
| Parameters for altitude | P_TOP, ZNU, ZNW, P, PB |
| | PH, PHB, PSFC |
| Wind components | U, V, W |
| $Q_0$ | HFX |
| $\overline{h}$ | PBLH |
| **CHIMERE model** | |
| Longitude, latitude | lon, lat |
| Altitude | hlay |
| Wind | winz, winm, winw |
| $Q_0$ | sshf |
| $\overline{h}$ | hght |

**Table 4.** *List of parameters read by the BACKPLUMES program to calculate trajectories.*

Parameters are the three-dimensional wind components, the boundary layer height $\overline{h}$, the surface sensible heat flux $Q_0$ and the altitude of each model layer. The wind components are used for the horizontal and vertical transport. The boundary layer height is used to define the vertical extent of the possible mixing and the surface sensible heat flux is used to know if the current modelled hour corresponds to a stable or unstable surface layer (for when the particle is close to the surface).

The first step of the calculation is to choose a starting point. The user has to select a date, longitude, latitude and altitude, obviously included in the modelled domain and during the modelled period. From this starting point, the model will calculate trajectories back in time. The number of trajectories is a up to the user and may be from one to several hundred of tracers.

In general, regional atmospheric models produce hourly outputs. But for some applications, an hourly time-step is not adapted. When the model mesh is very fine, the back-trajectories have no sense if they are calculated on an hourly basis. The air masses may cross several grid cells in 1 hour and a smaller integration time-step is required to preserve the trajectory continuity.





This is the problem to respect the Courant Freidrich Levy (CFL) number as in forward transport mode. In this program, the user can specify a sub-hourly step: the meteorological variables are then linearly interpolated between the two consecutive hours using a classical 1D linear interpolation. The altitude is not provided in the WRF output files. It is thus necessary to diagnose it. For all back-trajectories, note that all altitudes are Above Ground Level (AGL). The altitude is computed as follows:

$$p^* = p_{surf} - p_{top} \tag{14}$$

where $p_{surf}$ (PSFC) is the surface pressure and $p_{top}$ is the top pressure of the model domain. If $p_{top}$ is constant over the whole domain, $p_{surf}$ and thus $p^*$ are dependent on the horizontal grid cell.

$$z_0 = \frac{\Phi(1) + \Phi'(1)}{g} \tag{15}$$

where $\Phi$ is the geopotential (PHB) and $\Phi'$ (PH) its perturbation at vertical level $k$. $g$ is the acceleration of gravity, $g$=9.81 m s$^{-2}$. For each vertical level $k$, the layer thickness $\Delta z$ and the cell top altitude $z_k$ is estimated as:

$$
\begin{aligned}
d_m &= log\left(\frac{p^*\eta_M - p_{top}}{p^*\eta_M + p_{top}}\right) \\
d_u &= log\left(\frac{p^*\eta_M - p_{top}}{p^*\eta_F + p_{top}}\right) \\
z_1 &= \frac{\Phi(k) + \Phi'(k)}{g} \\
z_2 &= \frac{\Phi(k+1) + \Phi'(k+1)}{g} \\
\Delta z &= (z_2 - z_1)\frac{d_u}{d_m} \\
z(k) &= z_1 + \Delta z - z_0
\end{aligned}
\tag{16}
$$

where $\eta_M$ is eta values on full (w) levels (ZNW) and $\eta_F$ is eta values on half (mass) levels (ZNU). The layer thicknesses varying in space but also in time, this calculation is done for all trajectories and all time-step. At each time-step and for each trajectory, the position of the air mass is estimated by subtracting its pathway travelled as $\Delta\lambda$ and $\Delta\phi$ to the current position in longitude ($\lambda$) and latitude ($\phi$). To do so, all necessary variables are interpolated in 3D or 2D with our general interpolation program described in the previous section.





$$
\begin{aligned}
\phi_{rad} &= \phi\frac{\pi}{180} \\
\Delta x &= u\frac{3600}{\Delta t} \\
\Delta y &= v\frac{3600}{\Delta t} \\
\Delta\lambda &= \frac{\Delta x}{R\cos(\phi_{rad})}\frac{180}{\pi} \\
\Delta\phi &= \frac{\Delta y}{R}\frac{180}{\pi}
\end{aligned}
\tag{17}
$$

with the wind speed is provided in m s$^{-1}$ on an hourly basis, $R$ is the Earth radius as $R$=6371 km. The new position for one tracer is thus:

$$
\begin{aligned}
\lambda_{t-1} &= \lambda_t - \Delta\lambda \\
\phi_{t-1} &= \phi_t - \Delta\phi
\end{aligned}
\tag{18}
$$

The key point of this program is the choice made for the vertical mixing (Figure 5). Depending on the vertical position of the tracer, several hypotheses are made. Two parameters are checked for each tracer and each time-step: (i) the boundary layer height enables to know if the particle is in the boundary layer or above in the free troposphere, (ii) the surface sensible heat fluxes enables to know if the atmosphere is stable or unstable.

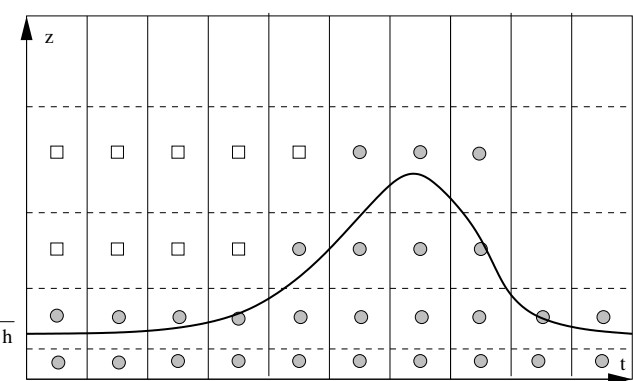

**Figure 5.** *Methodology for the vertical distribution as a function of the diurnal cycle of the boundary layer height.*

When the tracer is diagnosed to be in the boundary layer, there are two cases: the boundary layer is stable or unstable. If the boundary layer is stable, $Q_0 < 0$, the particle stays in the boundary layer at the same altitude. The new vertical position of the tracer is:

$$
z_{t-1} = z_t
\tag{19}
$$





If the boundary layer is unstable, $Q_0 > 0$, the particle is considered in the convective boundary layer and may be located at every level in this boundary layer the time before. Therefore a random function is applied to reproduce a potential vertical mixing.

$$z_{t-1} = Rand \times \overline{h} \tag{20}$$

It is considered that 15 mn is representative of a well-mixed convective layer. If the time step is larger than 15 mn, the random function is applied. But if the time step is lesser than 15 mn, the vertical mixing is reduced to the vicinity of the current position of the tracer. In this case, we have:

$$z_{t-1} = Rand \times \Delta z \times [z_t] \tag{21}$$

where $\Delta z = \frac{1}{2}(z_t^{k-1} + z_t^{k+1})$ and $k$ is the vertical model level corresponding to $z_t$. In the free troposphere, the evolution of the particle is considered to be influenced by the vertical wind component. A random function is applied to estimate its possible vertical motion with values between 0 and $w/2$ m s$^{-1}$. The vertical variability of the tracer's position in the free troposphere is calculated by diagnosing the vertical velocity as:

$$z_{t-1} \quad = \quad z_t - (0.5 + Rand)w\frac{3600}{\Delta t} \tag{22}$$

where *Rand* is a random value, between 0 and 1, and different for each tracer and each time.

### 7.2 Examples of back-trajectories computations

An example is presented for the same case and the WRF and CHIMERE models. The difference between the two models is the number of vertical levels. The online modelling system WRF-CHIMERE is used, meaning that the horizontal grid is the same (a large domain including Europe and Africa and with $\Delta x=\Delta y=60$km). The wind field is also the same, WRF sending this information to CHIMERE. The boundary layer height is different between the two models, WRF using the (Hong et al., 2006) schemes and CHIMERE using the (Troen and Mahrt, 1986) scheme. The surface sensible heat flux is the same between the two models, CHIMERE using the flux calculated by WRF. WRF has more vertical model levels than CHIMERE, thus meteorological fields are interpolated from WRF to CHIMERE. It impacts the horizontal and vertical wind fields.

Figure 6 presents the results of back-trajectories launched the 10 August 2013 at 12:00 UTC. The location is at longitude $+10^o$E and latitude $+25^o$N, altitude=0 m AGL. This location has no scientific interest but is in the middle of the domain, to have the longer as possible trajectories. The complete duration of trajectories represent 10 days back in time. A total of 120 trajectories are launched at the same position and time. They are randomly mixed when they are in the boundary layer to represent the mixing and the diffusion.





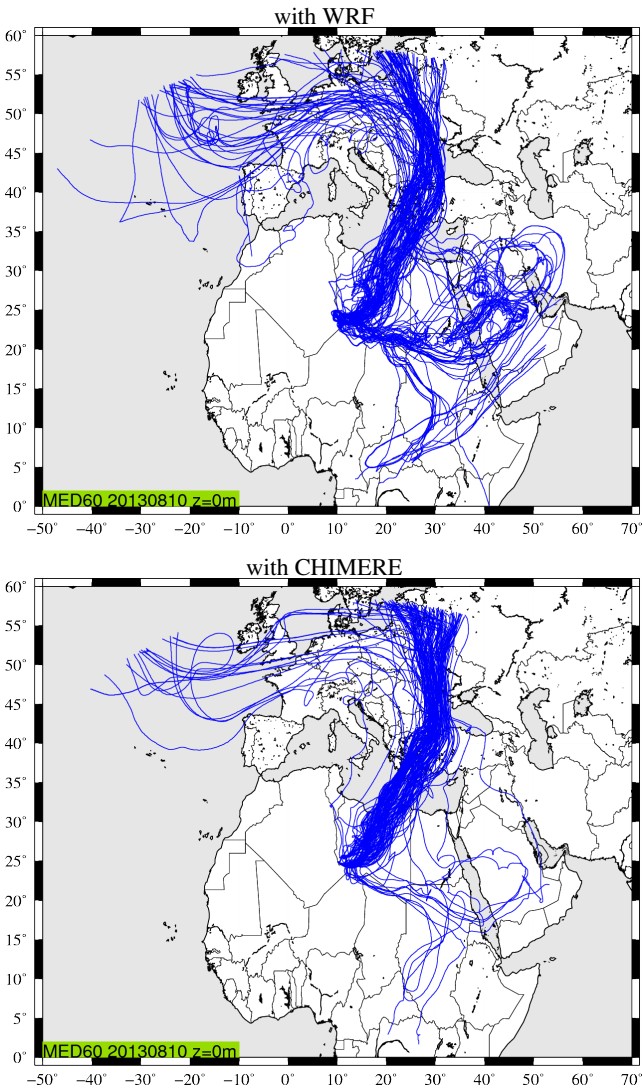

**Figure 6.** *Backplumes calculated using CHIMERE modelled meteorological fields. The starting point is at longitude $+10^{o}E$ and latitude $+25^{o}N$, altitude=0 m AGL and for the day 10 August 2013 at 12:00 UTC. It corresponds to a case studied during the CHARMEX campaign (Menut et al., 2015).*

The most important part of the plume comes from the North of the starting point. For this main plumes, the calculation is similar between the two models. Another large part of backplumes is modelled at the East of the starting point. However, this fraction is mainly modelled with WRF but not with CHIMERE where only a few trajectories are diagnosed. One possible explanation may be found by analyzing the vertical transport of the trajectories.



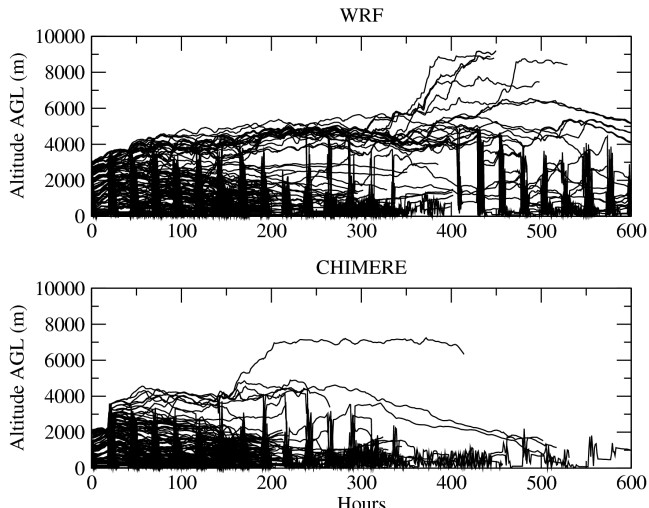

**Figure 7.** *Projection of all back-trajectories on a single time-altitude axis.*

Figure 7 presents all plumes displayed in the previous figure but projected along the same time-altitude axis. The differences between the two backplumes are mainly due at the calculation of the boundary layer height. When WRF diagnosed an altitude of $\approx 3000m$, CHIMERE diagnoses $\approx 2000m$, leading to different direction and wind speed, then to a split of the plumes with WRF but not with CHIMERE. This illustrates the sensitivity of the result to the forcing model. But, in both cases, the answer in our case is clearly that the main contribution of the air masses located at the starting point are mainly coming from the North.

## 8 Conclusions

A new interpolation program written in Fortran has been developed to interpolate on N-dimensional matrices. It has been evaluated for several dimension cases up to N=5. The code is fast compared to similar Python routines and highly portable in existing geophysical codes. The interpolation program works for any dimension N above 2 and designed to work with irregular but structured grids (characterized by a size for each dimension) or lookp-up tables. Already used in its "regular" version in CHIMERE, the "general" program has been tested on a new real application which calculates air mass back-trajectories from two widely used atmospheric models: CHIMERE and WRF. This interpolation program can be used for any application in Geophysics and Engineering Sciences but also to explore large structured matrices for Machine Learning applications.

*Code availability.* The current version of the models are freely available. The exact version of the model used to produce the results used in this paper is archived (i) on Zenodo for NterGeo at https://doi.org/10.5281/zenodo.3733278 under the GNU General Public License v3.0 or later, as are input data and scripts to run the model and produce the plots for all the simulations presented in this paper. The BACKPLUMES model is an open-source code and is available on the CHIMERE model web site https://www.lmd.polytechnique.fr/~menut/backplumes.php.





## Appendix A:  List of frequently used abbreviations

AGL         Above Ground Level

CFL         Courant Freidrich Levy

CHIMERE   National French CTM

CTM         Chemistry-transport model

CPU         Central Processing Unit

NMSE       Normalized Root Mean Square Error

PBL         Planetary Boundary Layer

PSFC        Surface Pressure

WRF         Weather Research and Forecasting model

## Appendix B:  Binary strategy

This piece of code shows the strategy to optimize the computation of weights for the "regular case". The idea is to minimize the number of operation to benefit from the calculation at each dimension. A non-optimized loop would require $2^N - 1$ multiplications while the optimized loop requires only $2^{N+1} - 4$ multiplications for the weight calculations. Then, for large values of $N \gg 2$, the ratio of required operations between the non-optimized and the optimized loop is $\approx (N-1)/2$.

```
! ...
nn=2**ndim                          ! ndim is the dimension of the case study
pn=2**(ndim−1)                      !
! Loop to convert k in binary
do k=0,nn−1
   do j=0,ndim−1
      if(btest(k,j)) then
           ibin(j,k)=1
      else
           ibin(j,k)=0
      endif
   enddo
enddo
...
! Main optimized loop (2 loops sequence) to calculate the weighting factors benefiting from
! the previous iteration on the main dimension i
do i=1,ndim                         ! Loop 1
  ni=2**i
  pi=ni/2
  do k=0,ni−1                        ! Loop 2
   delta=ibin(i−1,k)
   if(i.ne.1) then
```





```
     if(delta.eq.0) then
       ws(k)=weight(k)
       weight(k)=weight(k)*w(i,delta) ! where w is the weight on each dimension
     else
weight(k)=ws(k-pi)*w(i,delta)
     endif
     else
       weight(k)=w(i,delta)
     endif
enddo
    enddo
```

## Appendix C:  Code design

Note that *avedelta* and *maxdelta* arrays have been externalized to optimize the calculations. In the code package an independent program is available to calculate these arrays to be implemented in your main program. The program is written in Fortran *double precision* ingesting the following arguments:

```
    subroutine interpolation_general(&
    ndim,      &! Input  : Int
    maxdim,    &! Input  : Int
    kdim,      &! Input  : Array 1D, Int
vect,      &! Input  : Array 2D, Real
    vtable,    &! Input  : Array 1D, Real
    table,     &! Input  : Array 1D, Real
    avedelta,  &! Input  : Array 1D, Real
    maxdelta,  &! Input  : Array 1D, Real
table,     &! Input  : Array 1D, Real
    resu,      &! Output : Real
    inei,      &! Output : Integer
    neighbours,&! Output/Input : Array 2D, Int
    weights,   &! Output : Array 1D, Real
found      &! Output : Logical
    )
```

Some hard coded variables can be tested by the user to improve the results. They have been tested and some results are described in this paper. A recompilation is necessary if you change these values.

```
    logical , parameter          :: norm=.false.    ! Normalize or not by the average delta on each
dimension
    logical , parameter          :: verbose=.false. ! Level of message writing (.true. for debug)
    logical , parameter          :: neighb=.true.   ! .true. only find up to ndim+1 closest neighbours to
      be faster
                                                    ! .false. find up to 2\^NDIM closest neighbours
```





**Table C1.** Description of subroutine arguments

| Variable | Type | Description | Array dimension |
|---|---|---|---|
| $ndim$ | Integer | Dimension $N > 1$ | nd |
| $maxdim$ | Integer | Total number of elements of the input table $n = \prod_{i=1}^{N} I_i$ with $I_i$ the number of elements in each dimension $i$ | nd |
| $kdim$ | Integer 1D Array | Array of number of elements $I_i$ on each dimension $i$ | $(0:N)$ |
| $vect$ | Real 2D Array | Array storing in a 1 dimensional array the list of edges on each dimension | $(1:N, 1:n)$ |
| $vtable$ | Real 1D Array | Coordinate values of the point at which to interpolate data | $(1:N)$ |
| $table$ | Real 1D Array | Values for the list of known points $vect$ (input grid data) | $(1:n)$ |
| $avedelta$ | Real 1D Array | Inverse of average intervals on each dimension $N$ | $(N)$ |
| $maxdelta$ | Real 1D Array | Maximum intervals on each dimension $N$ | $(N)$ |
| $resu$ | Real | Result of interpolation for $vtable$ | nd |
| $inei$ | Integer | Number of neighbours | nd |
| $neighbours$ | Real 2D Array | Array of neighbours coordinates | $(1:2^N, 1:n)$ |
| $weights$ | Real 1D Array | Weight for each neighbour | $(1:2^N)$ |
| $found$ | Logical | Returns $true$ or $false$ if respectively the result is found or not found if the point is outside the bounds | nd |

```
integer , parameter          ::  iconf=1        ! Number of cell to account for before and after
                                                ! the closest point, iconf=2 can be tested not more
real(kind=iprec) , parameter ::  pnum=2.0d+00   ! p-distance parameter
```





**Appendix D: Irregular structured grid example in 5D**

Herebelow is an example of a 5D array input gridata with irregular intervals with the last dimension (5) depending on dimension
(1).

```
! Definition of main dimensions
isize=9                    !Outputgrid size
npoints=35                 !Inputgrid size
ndim=5                     !Dimension of the example
allocate(kdim(0:ndim))     !Number of element per dimension array
kdim(0)=1                  !Fake dimension for computation issues
kdim(1)=npoints
kdim(2)=npoints
kdim(3)=npoints
kdim(4)=npoints
kdim(5)=npoints
! Main array allocation
allocate(delta(ndim))
allocate(dstart(ndim))
allocate(vect(ndim,kdim(1),kdim(2),kdim(3),kdim(4),kdim(5)))
allocate(rando(ndim,1:npoints-1))                           !Random variable
maxdim=kdim(1)*kdim(2)*kdim(3)*kdim(4)*kdim(5)
!
! Definition of the input grid "vect" with delta, dstart as:
do j=1,ndim
  delta(j)=1.d+00/dfloat(npoints-1)/(dfloat(j)**3)
  dstart(j)=1.d+00/dfloat(npoints-1)/2.d+00/(dfloat(j)**3)
enddo
do j=1,ndim
  do i=1,npoints-1
    rando(j,i)=dfloat(int(dlog(rand()*10.d+00+1.d+00))+2)   !Random interval on each dimension
  enddo
  rando(j,:)=dfloat(npoints-1)*rando(j,:)/sum(rando(j,:))  !Normalisation of random intervals
enddo
vect=0.0d+00 ! Initialisation
! "vect" computation
do i=2,kdim(1)
  vect(1,i,:,:,:,:)= vect(1,i-1,:,:,:,:)+rando(1,i-1)*delta(1)
enddo
do j=2,kdim(2)
  vect(2,:,j,:,:,:)= vect(2,:,j-1,:,:,:)+rando(2,j-1)*delta(2)
enddo
do k=2,kdim(3)
  vect(3,:,:,k,:,:)= vect(3,:,:,k-1,:,:)+rando(3,k-1)*delta(3)
```





```
enddo
       do  l=2,kdim(4)
        vect(4,:,:,:,l,:)= vect(4,:,:,:,l-1,:)+rando(4,l-1)*delta(4)
       enddo
       do m=2,kdim(5)
do  i=1,kdim(1)
          vect(5,i,:,:,:,m)=vect(5,i,:,:,:,m-1)+rando(5,m-1)*delta(5)+&
                          &delta(5)*dfloat(i-1)/dfloat(kdim(1)-1) !Dim 5 depends on Dim. 1
        enddo
       enddo
!....
```



## Appendix E: Characteristics of Machine 1

- Architecture: x86_64

- CPU op-mode(s): 32-bit, 64-bit

- Byte Order: Little Endian

- CPU(s): 64

- On-line CPU(s) list: 0-63

- Thread(s) per core: 2

- Core(s) per socket: 8

- Socket(s): 4

- NUMA node(s): 8

- Vendor ID: AuthenticAMD

- CPU family: 21

- Model: 1

- Model name: AMD Opteron(TM) Processor 6276

- Stepping: 2

- CPU MHz: 2300.000

- CPU max MHz: 2300.0000

- CPU min MHz: 1400.0000

- BogoMIPS: 4599.83

- Virtualization: AMD-V

- L1d cache: 16K

- L1i cache: 64K

- L2 cache: 2048K

- L3 cache: 6144K





– Memory block size: 128M

    – Total online memory: 128G

    – Total offline memory: 0B

    – Linux version 3.10.0-1062.12.1.el7.x86_64 (mockbuild@kbuilder.bsys.centos.org) (gcc version 4.8.5 20150623 (Red Hat 4.8.5-39)

**Appendix F: Characteristics of Machine 2**

    – Architecture: x86_64

    – CPU op-mode(s): 32-bit, 64-bit

    – Byte Order: Little Endian

    – CPU(s): 96

– On-line CPU(s) list: 0-47

    – Off-line CPU(s) list: 48-95

    – Thread(s) per core: 1

    – Core(s) per socket: 24

    – Socket(s): 2

– NUMA node(s): 2

    – Vendor ID: GenuineIntel

    – CPU family: 6

    – Model: 85

    – Model name: Intel(R) Xeon(R) Platinum 8168 CPU @ 2.70GHz

– Stepping: 4

    – CPU MHz: 2701.000

    – CPU max MHz: 2701.0000

    – CPU min MHz: 1200.0000





- BogoMIPS: 5400.00

- Virtualization: VT-x

- L1d cache: 32K

- L1i cache: 32K

- L2 cache: 1024K

- L3 cache: 33792K

- NUMA node0 CPU(s): 0-23

- NUMA node1 CPU(s): 24-47

- Memory block size: 128M

- Total online memory: 190.8G

- Linux version 3.10.0-957.41.1.el7.x86_64 (mockbuild@x86-vm-26.build.eng.bos.redhat.com) (gcc version 4.8.5 20150623
(Red Hat 4.8.5-36)

*Author contributions.* Bertrand Bessagnet has developed the code. Laurent Menut and Bertrand Bessagnet has co-developed and imple-
mented the code in the BACKPLUMES.v2020r1 model. Maxime Beauchamp has supported Bertrand Bessagnet for the developments.

*Competing interests.* The author declares that there is no conflict of interest.

*Acknowledgements.* This research was funded by the DGA (French Directorate General of Armaments) grant number 2018 60 0074 in the
frame of the project NETDESA.



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
