# Peer review of "A N-dimensional Fortran Interpolation Program (NterGeo.v2020a) for Geophysics Sciences - Application to a back trajectory program (Backplumes.v2020r1) using CHIMERE or WRF outputs"

_Geoscientific Model Development, 2020_

## Referee Comment (RC1) · Anonymous Referee #1 · 6 Jul 2020

This manuscript presents an useful tool to support the scientific analysis of atmospheric model outputs. This tool allows the estimation of back trajectories of plumes and it is directly linked to commonly-used regional atmospheric and chemistry-transport models, such as WRF or CHIMERE. The fact that the tool is directly linked to these models allows a total consistence beteween forward and backward estimates, as the wind field and grid are the same in both cases. The methodology is well described, with a clear and well-structured overall presentation. The code is available, through a link provided

in the manuscript.

I strongly recommend the publication of this manuscript in GMD. Here some minor comments that I consider that could improve the manuscript (but not necessary for publication).

1. In page 36, the authors mention that BACKPLUMES is different than other back-trajectories models, such as Hysplit or Flexpart. Could the authors explain more the differences with the before-mentioned models? As the authors mention more processes than atmospheric motions, such as chemistry and deposition processes, can they be more precise, indicating which models consider those processes (further than only atmospheric motions)?

2. In the comparison with Hysplit, could the authors indicate if their methodology consider the same meteorological parameters?

3. It would be appreciated to include a comment (or to highlight if already included; apparently it is not included) about the target pollutants, if used for chemistry-transport models; if back trajectories are mainly estimated considering atmospheric motions this code can be used mainly for non-reactive pollutants.

4. The authors mention through the paper "particles". Please clarify this more (or if it is a general pollutant, not necessarily a particle)

5. Could it be possible (not necessary for publication) to have an example of the comparison with Hysplit and Python for the WRF and CHIMERE applications? It could be useful for potential users.

---

## Short Comment (SC1) · 15 Jul 2020

**A N-dimensional Fortran Interpolation Program (NterGeo.v2020a) for Geophysics Sciences - Application to a back-trajectory program (BACKPLUMES.v2020r1) using CHIMERE or WRF outputs**

Bessagnet, B., Menut, L., M.Beauchamps     5
https://www.geosci-model-dev-discuss.net/gmd-2019-337/

Dear Editor and reviewers,

We acknowledge the reviewers for the time spent to evaluate our work and for their minor revisions. We also acknowledge the Editor and we made all proposed changes in the revised manuscript. Please note that answers are in blue and after each reviewer's remark. When a large paragraph is added in the manscript, it    10 is here described in a grey box.

All reviewers remarks were taken into account and are detailed in this letter.

Text, references and Figures (captions and labels) were checked and corrected as requested.

Best regards,
Laurent Menut    15
July 15, 2020

**1 Reviewer #1**

This manuscript presents an useful tool to support the scientific analysis of atmospheric model outputs. This tool allows the estimation of back trajectories of plumes and it is directly linked to commonly-used regional atmospheric and chemistry-transport models, such as WRF or CHIMERE. The fact that the tool is directly linked to these models allows a total consistence beteween forward and backward estimates, as the wind field and grid are the same in both cases. The methodology is well described, with a clear and well-structured overall presentation. The code is available, through a link provided in the manuscript.

I strongly recommend the publication of this manuscript in GMD. Here some minor comments that I consider that could improve the manuscript (but not necessary for publication).

*Answer:*

We acknowledge the reviewer for these interesting comments. There is also some questions, certainly because our text was not clear enough. Some minor corrections are then put into the manuscript as suggested by the reviewer.

1. In page 36, the authors mention that BACKPLUMES is different than other back-trajectories models, such as Hysplit or Flexpart. Could the authors explain more the differences with the before-mentioned models? As the authors mention more processes than atmospheric motions, such as chemistry and deposition processes, can they be more precise, indicating which models consider those processes (further than only atmospheric motions)?

   *Answer:*

   There is no page 36 but we assume it is probably page 2 (introduction) or page 11. The paragraph describing the backplumes model was completely changed to answer these questions. The introduction is simplified (because it is not the place for a model description) and the 'backplumes' section is enriched with more details about the model. The new part in the introduction is:

   > In order to quantify the impact of such new interpolation program and show examples of its use, it is implemented in the back-tajectory model Backplumes, developed in the same team than the CHIMERE model, Mailler et al. (2017). This host model is well dedicated for this implementation, because the most important part of its calculation is an interpolation of a point in a model grid box.

   and the new paragraph for the presentation of the backplumes model is:

   > **7.1 The Backplumes model**
   > In order to test this new interpolation program, it is implemented in a backtrajectories model called "Backplumes". This model was already used in some studies such as (Mailler et al., 2016) and (Flamant et al., 2018) for example. Backplumes is open source and is available on the CHIMERE web site. Backplumes calculates backtrajectories from a starting point and a starting date. It is different from other 'backtrajectories' models, such as HYSPLIT (Stein et al., 2015), STILT (Lin et al., 2003), (Nehrkorn et al., 2010) and Flexpart (Pisso et al., 2019), because it is launching hundreds of passive tracers and plot as outputs all trajectories. Thus the answer is complementary compared to the other models: the output results is all possible trajectories, and not only the most probable.
   > An advantage of Backplumes for the WRF and CHIMERE users is that the code is dedicated to directly read output results of these models. Being developed by the CHIMERE developers teams, the code is completely homogeneous with CHIMERE in term of numerical libraries. Another advantage is that the code is very fast and calculates hundreds of trajectories in a few minutes. Using the wind fields of WRF or CHIMERE, and running on the same grid, the results of backtrajectories are fully consistent with the simulations done by the models.
   > The model is dedicated to calculate transport but not chemistry: only passive tracers are released. But a distinction could be made between gaseous or aerosol tracer: for the latter one,

> scavenging is calculated to have a more realistic trajectory. The model is easy to use and light because a small set of meteorological parameters is required. These meteorological parameters are described in Table **??** for WRF and CHIMERE.

2. In the comparison with Hysplit, could the authors indicate if their methodology consider the same meteorological parameters?

   *Answer:*
   Yes, the same meteorological parameters are used and it is now explain in the new 'backplumes' section.

3. It would be appreciated to include a comment (or to highlight if already included; apparently it is not included) about the target pollutants, if used for chemistry-transport models; if back trajectories are mainly estimated considering atmospheric motions this code can be used mainly for non-reactive pollutants.

   *Answer:*
   Backplumes can only calculates the transport of passive tracers. It was added in the new paragraph.

4. The authors mention through the paper "particles". Please clarify this more (or if it is a general pollutant, not necessarily a particle)

   *Answer:*
   Yes, it was corrected by 'tracer'.

5. Could it be possible (not necessary for publication) to have an example of the comparison with Hysplit and Python for the WRF and CHIMERE applications? It could be useful for potential users.

   *Answer:*
   There is no comparison with Hysplit because this is not the same kind of trajectories which are calculated. But there is a comparison of 'Backplumes' used with WRF and CHIMERE. There is an interest to compare the same kind of calculation with two different forcings since the goal of this paper is to present a new interpolation algorithm.

**References**

Flamant, C., Deroubaix, A., Chazette, P., Brito, J., Gaetani, M., Knippertz, P., Fink, A. H., de Coetlogon, G., Menut, L., Colomb, A., Denjean, C., Meynadier, R., Rosenberg, P., Dupuy, R., Dominutti, P., Duplissy, J., Bourrianne, T., Schwarzenboeck, A., Ramonet, M., and Totems, J.: Aerosol distribution in the northern Gulf of Guinea: local anthropogenic sources, long-range transport, and the role of coastal shallow circulations, Atmospheric Chemistry and Physics, 18, 12 363–12 389, https://doi.org/10.5194/acp-18-12363-2018, https://www.atmos-chem-phys.net/18/12363/2018/, 2018.

Lin, J. C., Gerbig, C., Wofsy, S. C., Andrews, A. E., Daube, B. C., Davis, K. J., and Grainger, C. A.: A near-field tool for simulating the upstream influence of atmospheric observations: The Stochastic Time-Inverted Lagrangian Transport (STILT) model, Journal of Geophysical Research: Atmospheres, 108, n/a–n/a, https://doi.org/10.1029/2002JD003161, http://dx.doi.org/10.1029/2002JD003161, 4493, 2003.

Mailler, S., Menut, L., di Sarra, A. G., Becagli, S., Di Iorio, T., Bessagnet, B., Briant, R., Formenti, P., Doussin, J.-F., Gómez-Amo, J. L., Mallet, M., Rea, G., Siour, G., Sferlazzo, D. M., Traversi, R., Udisti, R., and Turquety, S.: On the radiative impact of aerosols on photolysis rates: comparison of simulations and observations in the Lampedusa island during the ChArMEx/ADRIMED campaign, Atmospheric Chemistry and Physics, 16, 1219–1244, 2016.

Mailler, S., Menut, L., Khvorostyanov, D., Valari, M., Couvidat, F., Siour, G., Turquety, S., Briant, R., Tuccella, P., Bessagnet, B., Colette, A., Létinois, L., Markakis, K., and Meleux, F.: CHIMERE-2017: from urban to hemispheric chemistry-transport modeling, Geoscientific Model Development, 10, 2397–2423, 2017.

Nehrkorn, T., Eluszkiewicz, J., Wofsy, S., Lin, J., Gerbig, C., Longo, M., and Freitas, S.: Coupled weather research and forecasting-stochastic time-inverted lagrangian transport (WRF-STILT) model, Meteorology and Atmospheric Physics, 107, 51–64, https://doi.org/10.1007/s00703-010-0068-x, 2010.

Pisso, I., Sollum, E., Grythe, H., Kristiansen, N. I., Cassiani, M., Eckhardt, S., Arnold, D., Morton, D., Thompson, R. L., Groot Zwaaftink, C. D., Evangeliou, N., Sodemann, H., Haimberger, L., Henne, S., Brunner, D., Burkhart, J. F., Fouilloux, A., Brioude, J., Philipp, A., Seibert, P., and Stohl, A.: The Lagrangian particle dispersion model FLEXPART version 10.4, Geoscientific Model Development, 12, 4955–4997, https://doi.org/10.5194/gmd-12-4955-2019, https://www.geosci-model-dev.net/12/4955/2019/, 2019.

Stein, A. F., Draxler, R. R., Rolph, G. D., Stunder, B. J. B., Cohen, M. D., and Ngan, F.: NOAA's HYSPLIT Atmospheric Transport and Dispersion Modeling System, Bulletin of the American Meteorological Society, 96, 2059–2077, https://doi.org/10.1175/BAMS-D-14-00110.1, 2015.

---

## Referee Comment (RC2) · Anonymous Referee #2 · 20 Jul 2020

Review for https://www.geosci-model-dev-discuss.net/gmd-2020-88/gmd-2020-88.pdf

The manuscript "A N-dimensional Fortran Interpolation Program (NterGeo.v2020a) for Geophysics Sciences – Application to a back-trajectory program (BACK-PLUMES.v2020r1) using CHIMERE or WRF outputs" by Bertrand Bessagnet et al., describes an interpolation subroutine written in Fortran and its application to analysis of CHIMERE data. The topic lies withing the scope of GMD, but the manuscript in

the current state needs additional work in order to be considered for publication. The writing is not sufficiently clear, which makes the reading difficult. The current level of English is inadequate for publication in GMD. It needs to be revised and edited by a proficient speaker. It would be ideal if a native English speaker could language edit the manuscript.

Specific comments are listed below:

Title "A N-dimensional Fortran Interpolation Program (NterGeo.v2020a) for Geophysics Sciences - Application to a back-trajectory program (BACKPLUMES.v2020r1) using CHIMERE or WRF outputs" triggers a question. CHIMERE being an Eulerian off-line chemistry-transport model, it doesn't generate wind fields. Offline trajectory models require winds fields in order to run. In what sense BACKPLUMES uses CHIMERE output? CHIMERE outputs are mainly composition fields. Does it mean that BACK-PLUMES uses CHIMERE interpolated meteorological fields from WRF?

P1

Abstract

l4 "Fortran is a powerful and fast language, highly portable and easy to interface with other existing Fortran models." This phrase is problematic: Fortran is not a model. Most languages should allow interfacing code. Is the point that there are many existing geophysical models written in fortran? It could be replaced by something like: "Fortran is a fast and powerful language highly portable. It is easy to interface models written in Fortran with each other."

l5 "any Fortran compiler": did the authors tried all possible options? Why don't just say that is written in standard fortran and tested with two compilers? In fact there are KIND statements and those can be dialect dependant.

L5

l6 "novel optional parameter" This is unclear: novel with respect to what? Optional with

respect to what?

l6 "For the general program, the inverse distance is used for the weight calculation with a distance defined as a p-distance". This phrase is undefined at this point. Which weights? What is p? The abstract should be self contained.

l8 "Moreover, a real case of geophysics application embedding this interpolation program is provided and discussed" Is it a "geophysical application"?

l8 "it consists in determining back-trajectories using atmospheric dispersion or mesoscale meteorological model outputs, respectively from the widely used models CHIMERE and WRF." Here "atmospheric dispersion" is unclear when applied to CHIMERE, which as far as I know is an Eulerian chemistry model itself. "Dispersion" usually refers to Larangian models. The phrase reads as : "NterGeo.v2020a determines back-trajectories using atmospheric dispersion (?) from CHIMERE and meteorology from WRF." Is that the intended meaning?

l13 "used in its recent version V4.9.2 to propose horizontal and vertical interpolations" "used to propose interpolations" please revise the English expression.

l17 "full coupling strategy": Full coupling of what?

l18 Is ISORROPIA the chemistry and thermodynamics module? then please place the reference at the end of the phrase. Or it is an interpolation routine?

l19 "In this case" do you mean in such cases? Is this related to what is said above? If so please do not open a new paragraph which implies a new topic.

P2

l25 "In some studies," is too informal, please revise.

l31 (Scipy, 2014), unfortunately -> (Scipy, 2014),]. Unfortunately...

l32 "not enough optimized for our objective": Please make explicit which objective.

l33 "irregular (varying intervals) but structured" is unclear. What is the structure? Do you mean a square array?

l35 It is still unclear what the program precisely does and why the existing interpolation algorithms are not sufficient.

l36 "Atmospheric models (physics and/or chemistry) are commonly used in the Geophysics community." This phrase as it is contains little information. Mainly Lagrangian models are named later in the paragraph, maybe the authors mean "Lagrangian models (physics and/or chemistry) are commonly used in the Atmospheric Science community?

l39 It is "atmospheric motion" not "motions".

l40 "A new back-trajectory plume has been developed" do you mean "A new back-trajectory plume model has been developed"?

l41 spatial interpolation of what? Please specify.

l41 "the other codes": please specify which codes.

41 Unspecified: "some additional characteristics are implemented," please list the most important.

Message of section 1: The introduction suggests that the aim is to interpolate 3d aerosol data on trajectories calculated with BACKPLUMES.

Section 2)

l51 "The program supports": which program, NterGeo?

here it is still not completely clear what "irregular structured" mean. This may suggest an arbitrary triangular mesh. But it appears that what is meant is that the nodes are in multidimensional grid array with varying edge lengths, i.e., there is an N_1 x ... x N_n array of cells that are hexahedrons with arbitrary trapezoids as faces.

L57 "As it includes not specific options or function, version of a compiler, there is no reason to have limitations or errors with other compilers." Which compilers did the authors try? Better simply to say that "for portability, the code is written in standard Fortran without elements from any prticlular dielect."?

P3

L58 "The top shell calling script in the package provide two sets of options for ": provide*s* two sets of options...

l58 "Assuming the X array, the result of the function f transforming X to Y array in R can be expressed as: " is confusing, please rewrite. It is stated that Y is a scalar, not an array, otherwise express the suitable space Rˆm.

l62: Typography, please use "\verb||" or the verbatim environment and not italics for file names.

l63 "a classical bilinear interpolation" before you mentioned arbitrary dimensions, do you mean multilinear?

l 70 Style: please don't start a phrase with a symbol.

l 79 1-dimensionAL array

eq 5 please check and define all variables involved. Is the summation from k=1 or from j=1?

l 85 "closest neighbours" or vertices of containing grid cell? It may be that the closest vertices correspond to a neighbouring cell

l 94 an hard coded option -> A hard coded option

l 99 in a 1D array. -> into a 1D array.

l 125 "If the distance is too high compared to the characteristic" -> "If the distance is larger than the characteristic"...

l 141 "The time consuming is somehow proportional to the number of points in which to interpolate." This phrase is not grammatically correct and as stated does not add any important information. It is trivial that performing more operations takes longer time, the authors could replace "somehow" for something more precise.

l 143 "There is a discontinuity of the NMSE from p = 1 to p = 1+": why?

l 144 "The NMSE decreases with the number of points but a slight increase is observed 145 from $200 \times 200$ from $300 \times 300$." why?

153 This could certainly depends -> This could certainly depend

l 175 "can use output files": what kind of data? CHIMERE and WRF output countains a very large number of variables, which are necessary?

"This kind of model has mainly one calculation to do several times" please revise this phrase. It is difficult to make sense out of it. Do you mean: This kind of models (Lagrangian) contain a time loop that performs many times the same kind on calculation?

l178 "The first advantage of BACKPLUMES is to use the results of a simulation already performed. " do you mean that BACKPLUMES is an offline model?

l 179 "homogeneous with the "direct" simulation" do you mean consistent with the Eulerian forward simulation?

l 180 "BACKPLUMES is different than other back-trajectories models, such as Hysplit or Flexpart." In what is different?

Please correct the English: different than other -> different to other

l 181 "Since it is difficult to calculate correct probabilities back in time, the choice was made to randomly launch numerous trajectories and try to cover all possible origins." This phrase is very unclear, please clarify.

l 181 Flexpart's goal is not to "estimate the most likely trajectory as an envelope of

numerous possible trajectories."

enveloppe -> envelope

l 185: "The BACKPLUMES model is an open-source code and is available on the CHIMERE model web site." You should add a link with this statement. I looked in the website https://www.lmd.polytechnique.fr/chimere/ and I couldn't find any reference to BACKPLUMES.

186-189 This paragraph is useful but incolmplte. Since it is not described before (although it has been used in a couple of publications) you should specify better the numerics of the model.

192 " The number of trajectories is a up to the user and may be from one to several hundred of tracers.": "tracers" usually denote chemical species. Do you mean particles?

193: This paragraph starts with a discussion of the time step and the CFL condition and then jumps to the WRF specific calcultion of the z coordiante. It is difficult to follow the logic and has many English language errors. Please rewrite paying attention to the logical order in the definition of the variables for the equations.

202 "the horizontal grid cell" do you mean dependent on the horizontal coordinate?

Eq 15: does 1 correspond to the lowest level?

207 " is eta values on full (w) levels " please revise the English.

"The layer thicknesses varying in space but also in time, this calculation is done for all trajectories and all time-step. " please revise the English.

200-208: This section is very unclear. You have to better explain the rationale of the log pressure vertical interpolation and possibly move the technical details (maybe expanded) to an appendix.

Figure 5: "Methodology for the vertical distribution as a function of the diurnal cycle of the boundary layer height." Please define the circles, squares and the black thick line in the caption.

210-215: How do you know that this Euler scheme is accurate enough?

219-220 There is also mixing in a neutrally stable boundary layer, even in the absence of thermal convection. How do you treat such cases?

225 "Therefore a random function is applied..." Please define Rand.

228 "It is considered that 15 mn is representative of a well-mixed convective layer." Why? Can you add e.g. a reference?

233 "its possible vertical motion with values between 0 and w/2 m s−1." Where do these values come from?

237 Define Rand (as all symbols) the fist time is mentioned.

The treatment of mixing is very simplistic. The well mixed condition is not even mentioned. Although simplified approaches may find application in certain situations, is limitations and shortcomings should be better presented and discussed. There is abundant literature on Lagrangian methods in the atmopsphere to refer to.

239: "The difference between the two models is 240 the number of vertical levels." How many in each case?

241: "The wind field is also the same, WRF sending this information to CHIMERE." Please rewrite this phrase revising the English.

242: " WRF using the (Hong et al., 2006) schemes and CHIMERE using the (Troen and Mahrt, 1986) scheme." Are there many Hong et al. schemes? Please use \citet and not \citep in order to place the parenthesis in the right place.

245 Please use a proper LaTeX degree symbol.

259: " But, in both cases, the answer in our case is clearly that the main contribution of the air masses located at the starting point are mainly coming from the North." There is very little quantitative information in this statement.

267: "This interpolation program can be used for any application in Geophysics and Engineering Sciences but also to explore large structured matrices for Machine Learning applications." This conclusion is not supported by the main body of the manuscript. Especially the mention to Machine Learning. There are no results nor discussion of this topic. Please rework it in the discussion or remove from the conclusions.

---

## Author Response (AR1)

**A N-dimensional Fortran Interpolation Program (NterGeo.v2020a) for Geophysics Sciences - Application to a back-trajectory program (BACKPLUMES.v2020r1) using CHIMERE or WRF outputs**

Bessagnet, B., Menut, L., M.Beauchamps                                                              5
https://www.geosci-model-dev-discuss.net/gmd-2019-337/

Dear Editor and reviewers,

We acknowledge the reviewers for the time spent to evaluate our work and for their minor revisions. We also acknowledge the Editor and we made all proposed changes in the revised manuscript. Please note that answers are in blue and after each reviewer's remark. When a large paragraph is added in the manscript, it    10
is here described in a grey box.
All reviewers remarks were taken into account and are detailed in this letter.
Text, references and Figures (captions and labels) were checked and corrected as requested.

Best regards,
Laurent Menut    15
July 15, 2020

**1 Reviewer #1**

This manuscript presents an useful tool to support the scientific analysis of atmospheric model outputs. This tool allows the estimation of back trajectories of plumes and it is directly linked to commonly-used regional atmospheric and chemistry-transport models, such as WRF or CHIMERE. The fact that the tool is directly linked to these models allows a total consistence beteween forward and backward estimates, as the wind field and grid are the same in both cases. The methodology is well described, with a clear and well-structured overall presentation. The code is available, through a link provided in the manuscript. I strongly recommend the publication of this manuscript in GMD. Here some minor comments that I consider that could improve the manuscript (but not necessary for publication).

*Answer:*
We acknowledge the reviewer for these interesting comments. There is also some questions, certainly because our text was not clear enough. Some minor corrections are then put into the manuscript as suggested by the reviewer.

1. In page 36, the authors mention that BACKPLUMES is different than other back-trajectories models, such as Hysplit or Flexpart. Could the authors explain more the differences with the before-mentioned models? As the authors mention more processes than atmospheric motions, such as chemistry and deposition processes, can they be more precise, indicating which models consider those processes (further than only atmospheric motions)?

   *Answer:*
   There is no page 36 but we assume it is probably page 2 (introduction) or page 11. The paragraph describing the backplumes model was completely changed to answer these questions. The introduction is simplified (because it is not the place for a model description) and the 'backplumes' section is enriched with more details about the model. The new part in the introduction is:

   > In order to quantify the impact of such new interpolation program and show examples of its use, it is implemented in the back-tajectory model Backplumes, developed in the same team than the CHIMERE model, Mailler et al. (2017). This host model is well dedicated for this implementation, because the most important part of its calculation is an interpolation of a point in a model grid box.

   and the new paragraph for the presentation of the backplumes model is:

   > **7.1 The Backplumes model**
   > In order to test this new interpolation program, it is implemented in a backtrajectories model called "Backplumes". This model was already used in some studies such as (Mailler et al., 2016) and (Flamant et al., 2018) for example. Backplumes is open source and is available on the CHIMERE web site. Backplumes calculates backtrajectories from a starting point and a starting date. It is different from other 'backtrajectories' models, such as HYSPLIT (Stein et al., 2015), STILT (Lin et al., 2003), (Nehrkorn et al., 2010) and Flexpart (Pisso et al., 2019), because it is launching hundreds of passive tracers and plot as outputs all trajectories. Thus the answer is complementary compared to the other models: the output results is all possible trajectories, and not only the most probable.
   > An advantage of Backplumes for the WRF and CHIMERE users is that the code is dedicated to directly read output results of these models. Being developed by the CHIMERE developers teams, the code is completely homogeneous with CHIMERE in term of numerical libraries. Another advantage is that the code is very fast and calculates hundreds of trajectories in a few minutes. Using the wind fields of WRF or CHIMERE, and running on the same grid, the results of backtrajectories are fully consistent with the simulations done by the models.
   > The model is dedicated to calculate transport but not chemistry: only passive tracers are released. But a distinction could be made between gaseous or aerosol tracer: for the latter one,

> scavenging is calculated to have a more realistic trajectory. The model is easy to use and light because a small set of meteorological parameters is required. These meteorological parameters are described in Table **??** for WRF and CHIMERE.

2. In the comparison with Hysplit, could the authors indicate if their methodology consider the same meteorological parameters?

   *Answer:*
   Yes, the same meteorological parameters are used and it is now explain in the new 'backplumes' section.

3. It would be appreciated to include a comment (or to highlight if already included; apparently it is not included) about the target pollutants, if used for chemistry-transport models; if back trajectories are mainly estimated considering atmospheric motions this code can be used mainly for non-reactive pollutants.

   *Answer:*
   Backplumes can only calculates the transport of passive tracers. It was added in the new paragraph.

4. The authors mention through the paper "particles". Please clarify this more (or if it is a general pollutant, not necessarily a particle)

   *Answer:*
   Yes, it was corrected by 'tracer'.

5. Could it be possible (not necessary for publication) to have an example of the comparison with Hysplit and Python for the WRF and CHIMERE applications? It could be useful for potential users.

   *Answer:*
   There is no comparison with Hysplit because this is not the same kind of trajectories which are calculated. But there is a comparison of 'Backplumes' used with WRF and CHIMERE. There is an interest to compare the same kind of calculation with two different forcings since the goal of this paper is to present a new interpolation algorithm.

The manuscript "A N-dimensional Fortran Interpolation Program (NterGeo.v2020a) for Geophysics Sciences – Application to a back-trajectory program (BACK-PLUMES.v2020r1) using CHIMERE or WRF outputs" by Bertrand Bessagnet et al., describes an interpolation subroutine written in Fortran and its application to analysis of CHIMERE data. The topic lies withing the scope of GMD, but the manuscript in the current state needs additional work in order to be considered for publication. The writing is not sufficiently clear, which makes the reading difficult. The current level of English is inadequate for publication in GMD. It needs to be revised and edited by a proficient speaker. It would be ideal if a native English speaker could language edit the manuscript.

Specific comments are listed below:

Title "A N-dimensional Fortran Interpolation Program (NterGeo.v2020a) for Geophysics Sciences - Application to a back-trajectory program (BACKPLUMES.v2020r1) using CHIMERE or WRF outputs" triggers a question. CHIMERE being an Eulerian off-line chemistry-transport model, it doesn't generate wind fields. Offline trajectory models require winds fields in order to run. In what sense BACKPLUMES uses CHIMERE

output? CHIMERE outputs are mainly composition fields. Does it mean that BACK-PLUMES uses CHIMERE interpolated meteorological fields from WRF?

CHIMERE is off-line and can work with any meteorological model. CHIMERE requires wind fields. In most cases they come from WRF but they could come from IFS (ECMWF). Of course if CHIMERE uses WRF it will be better to directly uses WRF to avoid interpolation issues.

Abstract

L4 "Fortran is a powerful and fast language, highly portable and easy to interface with other existing Fortran models." This phrase is problematic: Fortran is not a model. Most languages should allow interfacing code. Is the point that there are many existing geophysical models written in Fortran? It could be replaced by something like: "Fortran is a fast and powerful language highly portable. It is easy to interface models written in Fortran with each other."

Yes you are right we reformulate according your suggestion.

L5 "any Fortran compiler": did the authors tried all possible options? Why don't just say that is written in standard Fortran and tested with two compilers? In fact there are KIND statements and those can be dialect dependent.

Right, we have changed to "*Our program does not need any libraries, it is written in standard Fortran and tested with two usual compilers.*"

L6 "novel optional parameter" This is unclear: novel with respect to what? Optional with respect to what?
We have changed to "A parameter (normalisation option) is provided…"

L6 "For the general program, the inverse distance is used for the weight calculation with a distance defined as a p-distance". This phrase is undefined at this point. Which weights? What is p? The abstract should be self contained.

We have decided to remove this technical sentence

L8 "Moreover, a real case of geophysics application embedding this interpolation program is provided and discussed" Is it a "geophysical application"?

Yes we have changed accordingly

L8 "it consists in determining back-trajectories using atmospheric dispersion or mesoscale meteorological model outputs, respectively from the widely used mod-els CHIMERE and WRF." Here "atmospheric dispersion" is unclear when applied to CHIMERE, which as far as I know is an Eulerian chemistry model itself. "Dispersion" usually refers to Larangian models. The phrase reads as : "NterGeo.v2020a deter-mines back-trajectories using atmospheric dispersion (?) from CHIMERE and meteorology from WRF." Is that the intended meaning?

Yes "Dispersion" is not appropriate. We have changed to "*it consists in determining back trajectories using chemistry transport or mesoscale meteorological model outputs*"

L13 "used in its recent version V4.9.2 to propose horizontal and vertical interpolations" "used to propose interpolations" please revise the English expression.

We have changed to "*is commonly used in its recent version V4.9.2 for horizontal and vertical interpolations to manage climate models outputs*"

L17 "full coupling strategy": Full coupling of what?

We changed to "*of a full coupling strategy between these modules*"

L18 Is ISORROPIA the chemistry and thermodynamics module? then please place the reference at the end of the phrase. Or it is an interpolation routine?

ISORROPIA is a chemistry and thermodynamics module. Yes we have moved the reference at the end of the sentence.

L19 "In this case" do you mean in such cases? Is this related to what is said above? If so please do not open a new paragraph which implies a new topic.

Yes the reviewer is right, we have corrected.

L25 "In some studies," is too informal, please revise.

We have removed "*In some studies,*"

L31 (Scipy, 2014), unfortunately -> (Scipy, 2014),]. Unfortunately...

Right, we have corrected.

L32 "not enough optimized for our objective": Please make explicit which objective.

We have changed to "Unfortunately this program is not really adapted to our problem, it could be not enough optimized for our objective as it can manage fully unstructured datasets."

L33 "irregular (varying intervals) but structured" is unclear. What is the structure? Do you mean a square array?

Yes, the difference is a bit tricky to explain but usually in the community of 3D modellers it is quite well apprehended. In 2D, "Structured" means you have the same number of grid point number on y direction for a given x value (idem for the other direction). "irregular" means you can have different delta x that can depend on y coordinates.

[Figure]

[Figure]

Structured Mesh          Unstructured Mesh

L35 It is still unclear what the program precisely does and why the existing interpolation algorithms are not sufficient.

We have added this sentence to clarify :"*In short, the novelty of this program is to fill the gap of interpolation issues between the treatment of very complex unstructured meshes and simple regular grid for a general dimension N.*"

L36 "Atmospheric models (physics and/or chemistry) are commonly used in the Geophysics community." This phrase as it is contains little information. Mainly Lagrangian models are named later in the paragraph, maybe the authors mean "Lagrangian models (physics and/or chemistry) are commonly used in the Atmospheric Science community?
L39 It is "atmospheric motion" not "motions".

This section has been rewritten according the suggestion of the previous reviewer and put later in the description of the back trajectory model.

L40 "A new back-trajectory plume has been developed" do you mean "A new back-trajectory plume model has been developed"?

L41 spatial interpolation of what? Please specify.

L41 "the other codes": please specify which codes.

L41 Unspecified: "some additional characteristics are implemented," please list the most important.

*For these 4 comments, it has completely been rewritten as : "In order to quantify the impact of such new interpolation program and show examples of its use, it is implemented in the back-trajectory model Backplumes, developed in the same team than the CHIMERE model (Mailler et al., 2017). This host model is well dedicated for this implementation, because the most important part of its calculation is an interpolation of a point in a model grid box. This paper describes (i) the methodology and the content of the interpolation program package NterGeo, and (ii) an application of this program embedded in the new "back-trajectory" program BACKPLUMES. These two codes are freely available (see code availability section)."*

Message of section 1: The introduction suggests that the aim is to interpolate 3d aerosol data on trajectories calculated with BACKPLUMES.

*No in fact as it is written "In order to quantify the impact of such new interpolation program and show examples of its use, it is implemented in the back-trajectory model Backplumes, developed in the same team than the CHIMERE model". However, the program can be used for any application that requires to interpolate.*

Section 2)

L51 "The program supports": which program, NterGeo?

*Yes we have clarified like this "The program NterGeo is fit for exploring irregular but structured grids or look-up tables defined by a unique size for each dimension which can be of course different from one to another dimension."*

here it is still not completely clear what "irregular structured" mean. This may suggest an arbitrary triangular mesh. But it appears that what is meant is that the nodes are in multidimensional grid array with varying edge lengths, i.e., there is an N_1 x ... x N_n array of cells that are hexahedrons with arbitrary trapezoids as faces.

*See my answer above regarding this point.*

L57 "As it includes not specific options or function, version of a compiler, there is no reason to have limitations or errors with other compilers." Which compilers did the authors try? Better simply to say that "for portability, the code is written in standard Fortran without elements from any prticlular dielect."?

*Yes we have changed to "The code does not need any libraries and is written in standard Fortran."*

L58 "The top shell calling script in the package provide two sets of options for ": provide*s* two sets of options...

*Corrected*

L58 "Assuming the X array, the result of the function f transforming X to Y array in R can be expressed as: " is confusing, please rewrite. It is stated that Y is a scalar, not an array, otherwise express the suitable space R^m.

*Corrected we have changed Y to an array structure.*

L62: Typography, please use "\verb||" or the verbatim environment and not italics for file names.

Corrected

L63 "a classical bilinear interpolation" before you mentioned arbitrary dimensions, do you mean multilinear?

Yes 'multilinear' is the right word.

L70 Style: please don't start a phrase with a symbol.

Yes we have corrected

L79 1-dimensionAL array

Corrected

Eq 5 please check and define all variables involved. Is the summation from k=1 or from j=1?

Corrected it was j=1

L85 "closest neighbours" or vertices of containing grid cell? It may be that the closest vertices correspond to a neighbouring cell

We have changed to "*closest vertice*"

L94 an hard coded option -> A hard coded option

Corrected

L99 in a 1D array. -> into a 1D array

Corrected

L125 "If the distance is too high compared to the characteristic" -> "If the distance is larger than the characteristic"...

Corrected

L141 "The time consuming is somehow proportional to the number of points in which to interpolate." This phrase is not grammatically correct and as stated does not add any important information. It is trivial that performing more operations takes longer time, the authors could replace "somehow" for something more precise.

We have changed to "As expected, the time consuming is obviously proportional to the number of points in which to interpolate."

L143 "There is a discontinuity of the NMSE from p = 1 to p = 1+": why?
I cannot explain this discontinuity that could be related to some rounding processes in fortran.

L144 "The NMSE decreases with the number of points but a slight increase is observed 145 from 200×200 from 300×300." why?

Increasing the number of points can produces noisy patterns, produce some instabilities that could impair the results.

L153 This could certainly depends -> This could certainly depend

Corrected

L175 "can use output files": what kind of data? CHIMERE and WRF output countains a very large number of variables, which are necessary?

This part has been rewritten thanks to reviewer 1 comments

"This kind of model has mainly one calculation to do several times" please revise this phrase. It is difficult to make sense out of it. Do you mean: This kind of models (Lagrangian) contain a time loop that performs many times the same kind on calculation?
This part has been rewritten thanks to reviewer 1 comments

L178 "The first advantage of BACKPLUMES is to use the results of a simulation already performed. " do you mean that BACKPLUMES is an offline model?

This part has been rewritten thanks to reviewer 1 comments as "*An advantage of Backplumes for the WRF and CHIMERE users is that the code is dedicated to directly read output results of these models. Being developed by the CHIMERE developers teams, the code is completely homogeneous with CHIMERE in term of numerical libraries. Another advantage is that the code is very fast and calculates hundreds of trajectories in a few minutes.*"
Yes BACKPLUME is offline, it can be run when CHMERE AND WRF outputs are ready.

L179 "homogeneous with the "direct" simulation" do you mean consistent with the Eulerian forward simulation?

No it refers to the way the code is written. It has been rewritten in a new section. See herebelow.

L180 "BACKPLUMES is different than other back-trajectories models, such as Hysplit or Flexpart." In what is different?

It has been changed to "In order to quantify the impact of such new interpolation program and show examples of its use, it is implemented in the back-trajectory model Backplumes, developed in the same team than the CHIMERE model, Mailler et al. (2017). This host model is well dedicated for this implementation, because the most important part of its calculation is an interpolation of a point in a model grid box."

Please correct the English: different than other -> different to other

L181 "Since it is difficult to calculate correct probabilities back in time, the choice was made to randomly launch numerous trajectories and try to cover all possible origins." This phrase is very unclear, please clarify.
L181 Flexpart's goal is not to "estimate the most likely trajectory as an envelope of numerous possible trajectories
enveloppe -> envelope

L185: "The BACKPLUMES model is an opensource code and is available on the CHIMERE model web site." You should add a link with this statement. I looked in the website https://www.lmd.polytechnique.fr/chimere/ and I couldn't find any reference to BACKPLUMES.

To answer all previous comments, the section has been completely rewritten as "*In order to test this new interpolation program, it is implemented in a backtrajectories model called "Backplumes". This model was already used in some studies such as (Mailler et al., 2016) and (Flamant et al., 2018) for example. Backplumes is open source and is available on the CHIMERE web site. Backplumes calculates backtrajectories from a starting point and a starting date. It is different from other 'backtrajectories' models, such as HYSPLIT (Stein etal.,2015), STILT(Linetal.,2003), (Nehrkornetal.,2010) and Flexpart (Pissoetal.,2019), because it is launching hundreds of passive tracers and plot as outputs all trajectories. Thus the answer is complementary compared to the other models: the output results is all possible trajectories, and not only the most probable. An advantage of Backplumes for the WRF and CHIMERE users is that the code is*

*dedicated to directly read output results of these models. Being developed by the CHIMERE developers teams, the code is completely homogeneous with CHIMERE in term of numerical libraries. Another advantage is that the code is very fast and calculates hundreds of trajectories in a few minutes. Using the wind fields of WRF or CHIMERE, and running on the same grid, the results of backtrajectories are fully consistent with the simulations done by the models. The model is dedicated to calculate transport but not chemistry: only passive tracers are released. But a distinction could be made between gaseous or aerosol tracer: for the latter one, scavenging is calculated to have a more realistic trajectory. The model is easy to use and light because a small set of meteorological parameters is required. These meteorological parameters are described in Table $$ for WRF and CHIMERE."*

L186-189 This paragraph is useful but incomplete. Since it is not described before (although it has been used in a couple of publications) you should specify better the numerics of the model.

This paragraph is just a summary of all useful variables. Then, their interest and use is explained later in the text (lines 215 for the vertical mixing for example). But, for clarity, the whole section was rewritten and a large part of explanations, not really mandatory for this paper about the interpolation routine, are put in Appendix.

L192 " The number of trajectories is a up to the user and may be from one to several hundred of tracers.": "tracers" usually denote chemical species. Do you mean particles?

Yes we have changed to "air particles" but also kept the concept of tracer.

L193 This paragraph starts with a discussion of the time step and the CFL condition and then jumps to the WRF specific calculation of the z coordinate. It is difficult to follow the logic and has many English language errors. Please rewrite paying attention to the logical order in the definition of the variables for the equations.

We have changed to "*In order to respect the Courant Friedrich Levy (CFL) number, a sub-time step may be calculated. If the input data are hourly provided (as in many regional models), the meteorological variables are interpolated between the two consecutive hours to obtain refined input data.*"

L202 "the horizontal grid cell" do you mean dependent on the horizontal coordinate?

Eq 15: does 1 correspond to the lowest level?

It corresponds to the ground level pressure. Then we have changed to "*on the first level grid*"

L207 " is eta values on full (w) levels " please revise the English.

We have changed to "where $\eta_{M}$ is its value on full (w) levels (ZNW) and $\eta_{F}$ is the eta value on half (mass) levels (ZNU)".

"The layer thicknesses varying in space but also in time, this calculation is done for all trajectories and all time-step. " please revise the English.

We have changed to "The layer thicknesses is space and time dependent, this calculation is performed for all trajectories and all time-step"

L200-208: This section is very unclear. You have to better explain the rationale of the log pressure vertical interpolation and possibly move the technical details (maybe expanded) to an appendix.

The use of pressure as vertical coordinate is classical in meteorological modeling. And when you have to interpolate between two altitudes, it is better to linearize the values using the log of the pressure before to use a linear interpolation. The reference to [Pielke, 1984] was added being the best way to see this point. Some sentences were added to clarify the text: the calculations are done in pressure vertical coordinates, but the input (the altitude of the starting point) and the outputs (the backplumes) are in

meters. Following the reviewer suggestion, some technical details are moved to an appendix.

Figure 5: "Methodology for the vertical distribution as a function of the diurnal cycle of the boundary layer height." Please define the circles, squares and the black thick line in the caption.

The Figure was removed because the explanation in the text is enough.

L210-215: How do you know that this Euler scheme is accurate enough?

The question is not clear

L219-220 There is also mixing in a neutrally stable boundary layer, even in the absence of thermal convection. How do you treat such cases?

This case is not treated. But it could be in a future version, the reviewer is right.

L225 "Therefore a random function is applied..." Please define Rand.

Yes, the definition is now in the manuscript

L228 "It is considered that 15 mn is representative of a well-mixed convective layer." Why? Can you add e.g. a reference?

Yes, it is in many articles about thermals and convective mixing. We prefer here to add the reference book explaining that, [Stull, 1988].

L233 "its possible vertical motion with values between 0 and w/2 m s−1." Where do these values come from?

Same reference, [Stull, 1988].

L237 Define Rand (as all symbols) the first time is mentioned.

Yes, it was done

The treatment of mixing is very simplistic. The well mixed condition is not even mentioned. Although simplified approaches may find application in certain situations, is limitations and shortcomings should be better presented and discussed. There is abundant literature on Lagrangian methods in the atmosphere to refer to.

We don't think it is simplistic: it is statistical. And it is the key point of this little backplumes model. It is only the assumption that, in a well convective mixed layer, you don't know at what altitude the tracer was within the boundary layer and 15mn ago.

L239 "The difference between the two models is 240 the number of vertical levels." How many in each case?

It depends on the user's choice when he made the simulation. But in our case, it was added in the text. But it is just an example, it is not very important regarding the goal of the paper

L241 "The wind field is also the same, WRF sending this information to CHIMERE." Please rewrite this phrase revising the English.

Ok, the sentence was replaced by: "The wind field is the same for the two models: CHIMERE using directly the wind field calculated by WRF."

L241 " WRF using the (Hong et al., 2006) schemes and CHIMERE using the (Troen and Mahrt, 1986) scheme." Are there many Hong et al. schemes? Please use \citet and not \citep in order to place the parenthesis in the right place.

Ok it was done

L245 Please use a proper LaTeX degree symbol.

$^{o}$ was replaced by $^{\circ}$ in the whole manuscript.

L259 But, in both cases, the answer in our case is clearly that the main contribution of the air masses located at the starting point are mainly coming from the North." There is very little quantitative information in this statement.

Yes, it is right. Mainly because the spread between the two models is not the main goal of this article. We do not want to present a geophysical analysis of the result but just illustrate the fact than the new interpolation scheme works well with two different models. However, the paragraph was updated to present more details with: "This illustrates the sensitivity of the result to the forcing model. But, in both cases, the answer in our case is clearly that the main contribution of the air masses located at the starting point are mainly coming from the North-East, crossing Tunisia, then the Mediterranean Sea and Europe. The main difference between the two calculations is the eastern part of the plume, more intense with WRF than CHIMERE."

L267 "This interpolation program can be used for any application in Geophysics and Engineering Sciences but also to explore large structured matrices for Machine Learning applications." This conclusion is not supported by the main body of the manuscript. Especially the mention to Machine Learning. There are no results nor discussion of this topic. Please rework it in the discussion or remove from the conclusions.

The reviewer is right we have removed the mention to Machine Learning.
* * *
Answers to Interactive comment on Geosci. Model Dev. Discuss., https://doi.org/10.5194/gmd-2020-88, 2020.

[revised manuscript text omitted]